# Maintenance of Methyl-Esterified Pectin Level in Pollen Mother-Cell Stages Is Required for Microspore Development

**DOI:** 10.3390/plants12081717

**Published:** 2023-04-20

**Authors:** Kazuya Hasegawa, Ai Ichikawa, Haruki Takeuchi, Atsuko Nakamura, Hiroaki Iwai

**Affiliations:** 1Institute of Life and Environmental Sciences, University of Tsukuba, Tsukuba 305-8572, Japan; 2Shizuoka Prefectural Research Institute of Agriculture and Forestry, Tea Research Center, Kikugawa 439-0003, Japan

**Keywords:** cell wall, pectin, pectin methylesterase, pollen, *Oryza sativa*

## Abstract

Pectin modification and degradation are vital for plant development, although the underlying mechanisms are still not well understood. Furthermore, reports on the function of pectin in early pollen development are limited. We generated *OsPME*-FOX rice lines with little methyl-esterified pectin even in the early-pollen mother-cell stage due to overexpression of the gene encoding pectin-methylesterase. Overexpression of *OsPME1* in rice increased the activity of PME, which decreased the degree of pectin methyl esterification in the cell wall. *OsPME1*-FOX grew normally and showed abnormal phenotypes in anther and pollen development, especially in terms of the pollen mother-cell stage. In addition, we examined modifications of cell-wall polysaccharides at the cellular level using antibodies against polysaccharides. Immunohistochemical staining using LM19 and LM20 showed that methyl-esterified pectin distribution and the pectin contents in pollen mother-cell wall decreased in *OsPME1*-FOX compared with the wild type. Thus, the maintenance of methyl-esterified pectin plays a role in degrading and maintaining the pollen mother-cell wall during microspore development.

## 1. Introduction

Pollen development is accompanied by a characteristic series of complex changes in the cell wall [1,2]. Immediately before meiosis, pollen mother cells produce callose within their cell walls, union by plasmodesmata between adjacent pollen mother cells, and the tapetum disappears. The cell walls between neighboring pollen mother cells are separated by degradation, and after cytoplasmic division, microspores are surrounded by callose, a cell wall called the prime exine. Later, in the tetrad stage, the cell wall is further degraded, and the microspores are released into the locule. At this stage, the tapetum disintegrates [1]. Thus, cell-wall degradation and synthesis, as well as modification, are highly regulated in microspore development (Appendix A).

The pollen wall is the outer structure of the pollen grain that protects the male gametophyte from physical and abiotic stresses. The pollen wall plays a vital role in pollen development, a key process in the life cycle of angiosperms [3]. The pollen wall consists of exine and intine and is elaborately organized. Intine is composed of pectin, structural proteins [4], and microfibril cellulose [5].

There are few reports on factors that specifically regulate changes in the cell-wall structure during pollen formation. The *dex1* mutant of *Arabidopsis thaliana* exhibits abnormal pollen formation and male sterility [6]. Another male sterility mutant, *ms33*, also exhibits abnormal intine formation [7]. There are many reports that the genes encoding cell-wall-modifying enzymes are expressed specifically in the pollen and anthers [8,9,10,11,12]. A recent comprehensive method identified many genes encoding enzymes involved in anther cell-wall modification [13,14]. More than 20% of the genes thus identified may be involved in pectin synthesis, degradation, and modification. This suggests that pectin is strongly involved in the pollen cell wall and its formation.

Structurally and functionally, pectin is the most complex polysaccharide in the plant cell wall [15]. Biochemically, pectins are polysaccharides that are rich in galacturonic acid. Pectins are divided into three main types: homogalacturonan, rhamnogalacturonan-I, and rhamnogalacturonan-II [15,16]. Homogalacturonan biosynthesis and modification have recently been identified as key determinants of plant development [15]. Pectin methylesterase de-methylates the C-6 of homogalacturonan, producing a freely negatively charged carboxyl group that catalyzes a reduced degree of the methyl esterification of pectin [17]. The de-methyl-esterified pectin molecules are either bound together by calcium ions to form supramolecular gels, such as egg-box structures [16], or they become targets for pectin-degrading enzymes such as polygalacturonases, which affect cell-wall texture and rigidity. Thus, pectin methylesterase (PME) plays an important role in pectin remodeling. Therefore, the de-methyl esterification of pectin by PME may affect pollen development by being involved in forming the wall. PME is ubiquitous throughout the plant lifecycle [17] and has important biological significance in plant growth and development, including pollen development, cell elongation, stem morphogenesis, cell attachment and separation, seed-coat mucus extrusion, and abiotic stress [18,19,20,21,22,23]. PME in higher plants is composed of two groups, depending on the presence or absence of a PME inhibitory domain (PMEI). Group 1 and Group 2 PMEs have a conserved PME domain (Pfam01095) with a specific and highly conserved amino acid fragment. Group 2 PMEs have an N-terminal extension called the PRO region, which has a PMEI domain (Pfam04043) [17]. In this study, we picked up *OsPME34* (Os11g0683800) from group 1 and *OsPME1* (Os01g0788400) from group 2 as rice PME genes that had characteristic expression patterns in floral organs in the database and investigated their overexpression and the effects of overexpression of these proteins.

Recent advances in our understanding of the PME function in reproductive development relate primarily to the role of PME in pollen formation and pollen tube growth as demonstrated using reverse genetics. Some PME genes are expressed specifically in pollen grains, such as NtPPME1 from *Nicotiana tabacum* [24,25]. In *Arabidopsis*, *QUARTET1* (*QRT1*), expressed in pollen and the surrounding anther tissue, has been shown to play a role in pollen-tetramer separation during flower development and to possess PME activity when expressed in *E. coli.* [26]. *QRT3*, which encodes the polygalacturonase found in *Arabidopsis*, acts in concert with *QRT1* and may lead to the degradation of de-methyl-esterified polygalacturonic acid in the primary wall of pollen mother cells.

The *Arabidopsis QRT1*, *QRT2*, and *QRT3* genes are required for normal pollen development [3,27]. In wild type (WT) plants, proper degradation of the cell wall of the pollen mother cell and subsequent microspore separation results in the release of individual microspores as single pollen grains. Loss of function of any of the three *QRT* genes prevents pollen-grain separation after meiosis and releases meiosis-associated mature pollen quads. The *qrt1* mutant properly deposits [27] and degrades [28] the callose cell wall of the pollen mother cell composed of callose, but the primary cell wall, including pectin of the pollen mother cell, remains after meiosis [28]. In WT, the callose wall and pectin wall were properly deposited and degraded. *QRT1* is a PME that functions at the tetrad stage. Pollen from *qrt2* mutants develops similarly but leaves mottled callose around the microspores [27]. Although these studies have greatly enhanced our understanding of the regulation of the pollen mother-cell wall, the regulation of pectin methyl esterification at the pollen mother-cell stage is still poorly understood.

We used OsPME-FOX rice with little methyl-esterified pectin even in the early-pollen mother-cell stage due to overexpression of the gene encoding PME, and immunohistochemical staining showed that methyl-esterified pectin distribution and pectin contents in the pollen mother-cell wall decreased in *OsPME1*-FOX compared with WT. From these results, we show that proper maintenance of methyl esterification level of pectin via PME is important for normal microspore development.

## 2. Results

### 2.1. OsPME1-FOX and OsPME34-FOX Overexpressed PME

For a systematic approach to analyze a gain-of-function phenotype, Hakata et al. [29] developed the FOX hunting system by using expression libraries for full-length cDNAs (fl-cDNAs) from rice at a maximum of 28,000 fl-cDNA clones in total, and individually overexpressed the fl-cDNAs in rice driven by the maize *ubiquitin-1* gene promoter. Among approximately 14,500 FOX rice lines, we identified two lines overexpressing fl-cDNAs for *OsPME1* and *OsPME34* and named them *OsPME1*-FOX and *OsPME34*-FOX. The *OsPME1* and *OsPME34* transcript levels were elevated in *OsPME1*-FOX and *OsPME34*-FOX, respectively (Figure 1A). PME activity in the flowers was also increased 5.2- and 4.1-fold in *OsPME1*-FOX and *OsPME34*-FOX, respectively, compared with that of the WT (Figure 1B).

### 2.2. Effect of Overexpression of PME1 and PME34 on the Pectin of the Cell Wall

The level of degree of pectin methyl esterification in the early stage of the flower was decreased to 27.2% and 69.7% in *OsPME1*-FOX and *OsPME34*-FOX, respectively, compared with the WT (Figure 2A). We also determined the uronic acid contents from polygalacturonase-soluble fraction of the cell wall of *OsPME1*-FOX and *OsPME34*-FOX. The pectin content in *OsPME1*-FOX and *OsPME34*-FOX was slightly decreased to 63.2% and 78.9%, respectively, compared to that of the WT (Figure 2B). The decrease in the level of degree of pectin methyl esterification was more significant in *OsPME1*-FOX than in *OsPME34*-FOX.

### 2.3. OsPME1-FOX and OsPME34-FOX Showed Abnormal Anther Development

*OsPME1*-FOX and *OsPME34*-FOX and control plants showed no differences in vegetative tissue (Figure 3). *OsPME*-FOX1 and *OsPME34*-FOX plants showed various abnormal phenotypes in the male reproductive organs, but no differences from WT were observed in the female reproductive organs (Figure 4A,F). In male reproductive organs, anthers were curled and short with a white color in *OsPME1*-FOX plants compared to WT (Figure 4B,C). Anthers in *OsPME34*-FOX plants were slightly shorter than in WT (Figure 4G, H). Furthermore, in *OsPME*-FOX1, it was observed that few pollen grains were present in the anthers (Figure 4E) compared to WT (Figure 4D). However, in *OsPME34*-FOX, pollen grains were similar to WT (Figure 4I,J). During reproductive development, the fertility rate decreased by ~12% in *OsPME1*-FOX and ~26%*OsPME34*-FOX. In the *OsPME1*-FOX phenotype, the decrease in the degree of pectin methyl esterification level was significant (Figure 2A), and abnormal anther development could be observed more stably than that in *OsPME34*-FOX; thus, detailed microscopy experiments were performed using *OsPME1*-FOX.

### 2.4. OsPME1-FOX Showed Abnormal Pollen Development

To determine the stage at which pollen development is disrupted in *OsPME1*-FOX, thin sections were obtained from the flowers of WT and *OsPME1*-FOX plants at various stages [14,30] (Figure 5). We divided rice-anther development after the pollen mother-cell stage into five stages. First, in the stage where pollen mother cells are observed in WT (Figure 5A,B). In *OsPME1*-FOX, pollen mother cells stopped developing in the early-pollen mother-cell stage (Figure 5F,G), and aberrant cells enlarged and filled the anthers where pollen mother cells are normally developed (Figure 5G–I). In WT, tapetum was developed at the tetrad stage and mitosis stage (Figure 5C,D), whereas tapetum morphogenesis could not be observed in the anther of *OsPME1*-FOX (Figure 5H,I). Observation of anthers in the mature stage showed that mature pollen was formed in WT (Figure 5E), whereas no pollen was observed in *OsPME1*-FOX (Figure 5J).

### 2.5. OsPME1-FOX Showed Abnormal Distribution of Methyl-Esterified and De-Methyl-Esterified Pectin in Pollen Mother-Cell Stage of Anther

Since the defect responsible for low fertility was expected to be an abnormality in the methyl-esterification condition of pectin at the pollen mother-cell stage, which has abnormal morphogenesis in *OsPME1*-FOX, we prepared cross-sections of anthers from the early- and late-pollen mother-cell stages of *OsPME1*-FOX and WT plants by staining with a de-methyl-esterified pectin-specific antibody (LM19) and a methyl-esterified pectin-specific antibody (LM20) (Figure 6 and Figure 7). In WT, signals for both de-methyl-esterified and methyl-esterified pectin were observed in anthers at the early pollen mother-cell stage (Figure 6B,F), while in anthers at the late pollen mother-cell stage, the methyl-esterified pectin signal was very weak and the signal of the dimethyl-esterified pectin was stronger than in the previous stage (Figure 7B,F). On the other hand, in anthers at the early pollen mother-cell stage of *OsPME1*-FOX, the signal of de-methyl-esterified pectin could be observed (Figure 6J), but the signal of methyl-esterified pectin was not detected (Figure 6N). Although methyl-esterified pectin is also included in the Golgi vesicles, these experiments were focused on the cell wall, exposure times were the same in WT and *OsPME1*-FOX, and the amount of pectin itself was much higher in the cell wall than in the Golgi vesicles, and therefore methyl-esterified pectin signals in Golgi vesicle are very difficult to observe. In anthers at the late-pollen mother-cell stage of *OsPME1*-FOX, and de-methyl-esterified pectin and methyl-esterified pectin were hardly detected (Figure 7J,N). Green autofluorescence signals for sporopollenin, the main component of exine, were not detected in WT and *OsPME1*-FOX in the early- and late-pollen mother-cell stages (Figure 6D,H,L,P and Figure 7D,H,L,P).

## 3. Discussion

During pollen development, the cell walls of many tissues changes, including the tapetum, pollen mother cell, vesicle, and transmitting tissue of pistil [31,32,33,34]. A lot of enzymes for pectin degradation and modification have been reported from the anthers of petunia, tobacco, and *Arabidopsis* [8,9,10,11,32]. These enzymes include polygalacturonases, which directly depolymerize the pectin, as well as PME.

PME plays an important role in plant-cell separation and cell-wall modification. Primary cell-wall pectins are composed of the polysaccharides homogalacturonan, rhamnogalacturonan-I, and rhamnogalacturonan-II. The main component of all three types of pectin is galacturonic acid. When pectin is synthesized in Golgi body, galacturonic acid residues are methyl-esterified by pectin methyltransferase and released into the cell wall as methyl-esterified pectin [33,34]. Pectin chains may need to be depolymerized when cell-wall expansion is required for growth or when degradation progresses, as in fruit softening [35]. Depolymerization of pectin chains is initiated by de-methyl esterification by PME. The resulting de-methylated pectin is then depolymerized and degraded by polygalacturonase. Hence, pectin, in which methyl groups are removed by PME, is known to be a target of polygalacturonase. Therefore, the regulation of pectin methyl esterification by PME is involved in the control of pectin degradation as well as the control of cell-wall properties by binding to calcium.

The overexpression of *OsPME1*-FOX and *OsPME34*-FOX resulted in increased PME activity (Figure 1) and a reduced degree of methyl esterification of pectin (Figure 2). On the other hand, the amount of pectin in *OsPME1*-FOX and *OsPME34*-FOX was slightly lower than in WT (Figure 2), even though PG activity was not increased compared to WT (data not shown). It is possible that many de-methyl-esterified pectins in *OsPME1*-FOX and *OsPME34*-FOX are easy targets for polygalacturonase and are therefore also susceptible to degradation. In the *OsPME1*-FOX phenotype, the decrease of the degree of pectin methyl esterification level and deficient development in pollen deficient could be observed more stably than in *OsPME34*-FOX. *OsPME1* has a conserved PME domain and PMEI domain (group 2 PME), and *OsPME34* only has PME domain (group 1 PME). Group 2 PME may affects pollen development than group 1 PME.

Our results showed that *OsPME1*-FOX and *OsPME34*-FOX affect normal pollen development (Figure 4, Figure 5, Figure 6 and Figure 7). There are many reports of mutants with abnormal pollen development. For example, *SPL/NZZ*, *BAM1*, *2*, *EMS1/EXS*, *SERK1*, *2*, *TPD1* and *OsGRP2* all function at an early stage and determine tapetal formation [4,36,37,38,39,40,41]. Thus, cell-wall development in tapetum is often required for pollen development. On the other hand, the cell-wall development of the stage of the pollen mother cell is also very important for pollen development. The *bam1* and *bam2* double mutant displays developmental defects at the early anther stage and lacks somatic cell layers, which suggests that these genes promote cell division and differentiation. In this study, we showed that the ubiquitously high expression of PMEs causes significant changes in pollen development with little pollen development (Figure 4 and Figure 5). This is due to the lack of methyl-esterified pectin distribution of the cell wall in early pollen mother cells (Figure 6), suggesting that overexpression of *OsPME1* probably leads to the progressive degradation of pectin in the early stages of pollen mother-cell development (Figure 5), and thus fails to maintain sufficient pectin levels (Figure 7). PME catalyzes the de-methyl esterification of pectin and can have distinctly different physiological consequences depending on the pattern of de-methyl esterification within the pectin [18]. When de-methyl esterification occurs randomly, protons are released, promoting polygalacturonase activity, and degradation of the cell-wall pectin. On the other hand, de-methyl esterification of pectin residues may occur linearly. As a result, large blocks of negatively charged pectin residues are thought to interact with calcium ions to form pectate-Ca gels and stiffen the cell wall [25]. The mode of de-methyl esterification of a particular PME has been shown to be influenced by many factors, including pH, ion concentration, and the existing state of pectin methyl esterification [42]. In the present study, pectin levels were clearly reduced in *OsPME1*-FOX (Figure 7). This strongly suggests that the effect of *OsPME* was not the formation of the pectin-Ca gel, but rather the progressive degradation of pectin by polygalacturonase. Our results suggest that it is important to maintain a high degree of pectin methyl esterification in the early stages of pollen mother-cell development through appropriate pectin methyl esterification regulation via PME. Additionally, the distribution of methyl-esterified pectin was not detected in WT at the later stages of pollen mother cell development (Figure 7). In *QRT3*, which encodes a polygalacturonase that degrades de-methyl-esterified pectin in the cell wall of pollen mother cells, loss of function prevents pollen development [3]. This suggests that maintaining a low degree of methyl esterification of pectin in late pollen mother cells and enriching the pollen mother-cell wall with de-methyl-esterified pectin may support *QRT3* function.

Overexpression of PME also did not affect the development of the vegetative organ (Figure 3). Monocotyledonous plants, including rice, have been reported to have very low pectin content in their vegetative organs. Overexpression of polygalacturonase also did not affect development of the vegetative organ [43], suggesting that pectin may not be very important for vegetative organ development in rice.

From our results and these studies, we conclude that maintaining the high degree of pectin methyl esterification in the cell wall during early pollen mother cells is directly and specifically involved in the degradation of pectic polysaccharides in the pollen mother-cell wall at the appropriate timing (Figure 8). In the *Arabidopsis qrt* mutant, microspores do not separate and remain attached to the characteristic tetrahedral aggregate of four pollen grains [27]. This separation failure suggests that the pectin component of the cell wall of the pollen mother cell remains around the microspores after callose degradation [28]. It has been suggested that modification and degradation of the pectin component of the microspore primary cell wall within the cell wall of the pollen mother cell are important for the separation between developing microspores. Ectopic overexpression of PME improperly degrades the pectin in the pollen mother-cell wall, presumably causing the developing pollen grains to become mechanically constrained and remain strongly attached to the pollen grains (Figure 8). *OsPME1*-FOX were mixed and abnormal cells with fused outer-cell layers were observed. This suggested that this leads to the inter-digitation of the polysaccharides of the developing cell walls of the pollen grains, resulting in the fusion of the exine layers at the point of oppression. Intermixing of cell-wall polysaccharides leading to abnormal cell fusion has previously been invoked to explain the phenotype of the fiddlehead and other mutants [44]. These results provide a partial answer to the question of why plants have numerous enzymes for pectin methyl esterification [45]. This is the first report on the regulation of pectin methyl esterification in the cell wall of pollen mother cells during microspore development.

## 4. Materials and Methods

### 4.1. Plant Material and Growth Conditions

Rice plants of the WT (Oryza sativa cv. Nipponbare) and FOX lines, which carry overexpression constructs for *OsPME1* (Os01g0788400) and *OsPME34* (Os11g0683800), respectively, were used. The plants were grown in soil in a greenhouse during the natural growing season at 28 °C under 1600 µmol s–1 m–2illumination (normal light condition), and in a growth chamber at 28 °C under 115 µmol s–1 m–2 illumination (weak light condition). The presence of the constructs in genomic DNA was confirmed by PCR using T3 generation plants. Transgenic lines were selected on hygromycin-containing agar and tested regarding the heritability of the expression pattern and altered sugar traits. To analyze gain-of-function phenotypes, Nakamura et al. (2007) and Hakata et al. (2010). developed the FOX hunting system using fl-cDNA expression libraries from rice containing 28,000 fl-cDNA clones and overexpressed the fl-cDNAs in rice under the control of the maize ubiquitin-1 promoter. Among approximately 14,500 FOX rice lines, two overexpressed fl-cDNAs for *OsPME1* (Os01g0788400) and *OsPME34* (Os11g0683800); these were named *OsPME1*-FOX and *OsPME34-*FOX, respectively.

### 4.2. Monitoring of Anther Development

The developmental stages of the rice pollen were defined in the same methods described in Itoh et al. [30] and Fujita et al. [14], based on anther length. Spikelets of different developmental stages were collected, fixed in PFA, and dehydrated through an ethanol series. The samples were embedded in Technovit 7100 resin (Hereaus Kulzer) and polymerized at room temperature. Transverse sections of 3 µm were cut using a Leica VT1200S (Leica Microsystems, Wetzlar, Germany) and stained with 0.1% (*w*/*v*) toluidine blue O in distilled water (DW). The sections were visualized by microscopy (DMRB, Leica Microsystems, Wetzlar, Germany).

### 4.3. Extraction of RNA and Analysis of Gene Expression

Plant material was frozen in liquid nitrogen and ground using a Tissue Lyser II (Qiagen, Hilden, Germany). The following procedure is based on the method described by Hyodo et al. [35]. Flowers with anthers less than 1 mm in length from *OsPME1*-FOX and *OsPME34*-FOX were sampled. Total RNA was extracted using an RNeasy Plant Mini Kit (Qiagen) and DNase I (Roche, Basel, Switzerland) according to the manufacturer’s instructions. cDNA was synthesized using ReverTra Ace^®^ (Toyobo, Tokyo, Japan), as specified by the manufacturer. For the *OsPME1*-FOX line, transcripts were quantified using the primers *OsPME1*-forward (5′-GAAGCAGTTCCCGACGTT-3′) and *OsPME1*-reverse (5′-CGCTCTGGTCCGTGATGA-3′). For the *OsPME34*-FOX line, transcripts were quantified using the primers *OsPME34*-forward (5′-GGCCTCCACTACATCAAGGA-3′) and *OsPME34*-reverse (5′-TGACGCAGTGGAATTACTCG-3′). As an endogenous control, 17S rRNA transcript was quantified using the primers 17S rRNA-forward (5′-GCAAATTACCCAATCCTGAC-3′) and 17S rRNA-reverse (5′-CTATTGGAGCTGGAATTACC-3′). The products were separated in a 2% agarose gel and stained with ethidium bromide. Quantitative reverse transcription–PCR (qRT–PCR) analysis was performed using SYBR Green I (Qiagen) with cDNA as the template on a Model 7000 Sequence Detection System (Applied Biosystems, Foster City, CA, USA). The ubiquitin gene (EU604080) was used as the normalizing reference gene.

### 4.4. Determination of Pectin Methylesterase Activity

Pectin methylesterase activity was assayed with a continuous spectrophotometric method according to Hyodo et al. [35]. Inflorescence, including anthers less than 1 mm in length from *OsPME1*-FOX and *OsPME34*-FOX, were sampled. Activity measurements were conducted at 20 °C and pH 7.5 in a cuvette containing 2 mL pectin, 0.15 mL bromothymol blue (BTB) and 0.55 mL distilled water. An addition of 0.3 mL enzyme extract started the reaction, and the residual enzyme activity was immediately assayed. The change in absorbance at 620 nm was recorded for 10 min in a UV/VIS spectrophotometer (Perkin-Elmer, Waltham, MA, USA). The activity values reported are an average of three independent measurements.

### 4.5. Extraction and Analysis of Cell-Wall Polysaccharides

The cell-wall extraction and analysis were conducted in accordance with Sumiyoshi et al. [46], with slight modifications. Rice inflorescences were frozen in liquid nitrogen and ground using a Tissue Lyser II (Qiagen) at 30 Hz for 2 min. A methanol/chloroform mixture (1 mL, 1:1, *v*/*v*) was added to the samples, followed by centrifugation at 15,000 rpm for 5 min and supernatant removal; the process was repeated twice. After the last supernatant removal, the samples were air-dried, and the resulting alcohol-insoluble residue (AIR, dry cell wall) was used as cell-wall material. The dry cell walls were extracted with 50 mM Na_2_CO_3_ at room temperature for 2 h as pectic fractions.

### 4.6. Determination of Uronic Acid

Uronic acid was determined using the method of Blumenkrantz and Gustav (1973). Briefly, 1 mL ion-exchange water was added to each sample, and 1 mL iced concentrated sulfuric acid (0.025 M borax) was mixed into 200 µL of each sample. After heating in 100 °C water for 10 min and cooling in ice, 40 µL carbazole solution (125 mg carbazole, 100 mL ethanol) was mixed into each sample. The samples were heated in 100 °C water for 15 min, then cooled in ice; absorbance was measured at 530 nm (GENESIS 10S UV-VIS; Thermo Scientific). The cell-wall material was treated for 4 h at 4 °C with 0.1 M NaOH to saponify the methyl and acetyl esters. The suspensions were adjusted to pH 5.0 with 10% (vol/vol) glacial acetic acid and then treated for 16 h at 30 °C with a homogeneous preparation of endo-polygalacturonase (EPG) from *Aspergillus niger* [2.5 units, Megazyme, Wicklow, Ireland; 1-unit releases 1 μmol of reducing sugar min^−1^ from a 1% (weight/volume) solution of polygalacturonate at pH 5.0 and 25 °C. The suspensions were centrifuged, and the insoluble residues were washed with water.

### 4.7. Determination of the Pectin Methyl Ester Content

The methyl ester group was determined quantitatively by an enzymatic method involving an alcohol oxidase/formaldehyde dehydrogenase system. For hydrolysis of methyl esters bound to pectin, 0.1 m KOH (100 µL) was added to the pectin fraction (100 µg/100 µL), followed by standing for 1 h at room temperature. The methanol released was determined. The reaction mixture, composed of 100 mM glutathione (60 µL), 100 mM NAD^+^ (60 µL), alcohol oxidase (1 unit) and FADH (2 units) in 0.2 M potassium phosphate buffer (pH 7.5) in a total volume of 2.9 mL, was placed in screw-cap tubes. Aliquots (100 µL) of methanol standards (0.5–10 µg) or the pectin hydrolysates containing 50 pg of galacturonic acid were added to the tubes. The tubes were incubated at 25 °C for 30 min. The methanol content was calculated using e = 6.2 × 10^3^ mol^–1^ cm^–1^ for NADH at 340 nm. The degree of methyl esterification was expressed as the molar per cent of methyl ester groups per d-galacturonic acid residues.

### 4.8. Immunohistochemistry

The following procedure is based on the method described by Hasegawa et al. [33]. Samples were fixed in 4% paraformaldehyde, 0.25% glutaraldehyde and 0.05 M phosphate buffer (pH 7.5). Transverse 3 µm sections were cut using a Leica RM2145 microtome (Leica Microsystems) and stained with 0.1% (*w*/*v*) toluidine blue O in distilled water. For immunohistochemistry, sections were subjected to immunohistochemical analysis using TSA Kit #12 (Invitrogen, Eugene, OR, USA). Primary antibodies for LM19 and LM20 (PlantProbes, Leeds, UK) were used at a dilution of 1:20. Negative controls lacked the primary antibody. Sections were visualized with fluorescence microscopy (Leica; DMRB).

### 4.9. Statistical Analysis

The data were expressed as the mean values ± SD taken from 4–9 independent biological experiments. The experimental data of the samples were statistically analyzed through one-way analysis of variance (ANOVA) with Tukey’s post hoc test using Statistica 13.1 software (StatSoft, Inc., Tulsa, OK, USA). The results with a *p*-value ≤ 0.05 and a *p*-value ≤ 0.01 were considered statistically significant.

## Figures and Tables

**Figure 1 plants-12-01717-f001:**
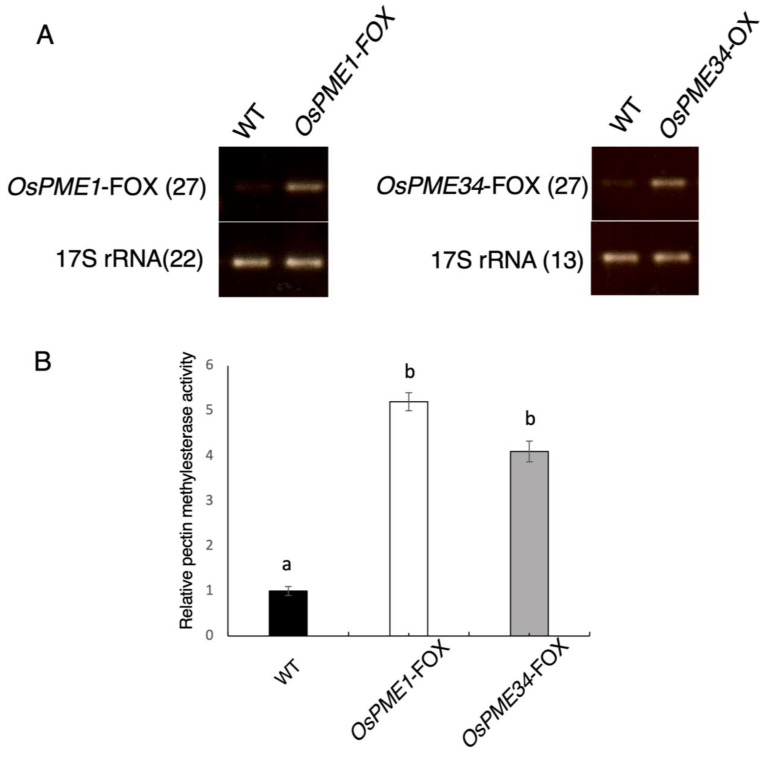
Characteristics of the *OsPME1*-FOX and *OsPME34*-FOX lines. (**A**) RT–PCR analysis of inflorescence transcripts from WT, *OsPME1*-FOX and *OsPME34*-FOX lines. The expression levels of *OsPME1*-FOX and *OsPME34*-FOX were higher than in the WT. The numbers in parentheses show the numbers of PCR cycles. These analyses were performed at least three times with similar results. (**B**) Pectin methylesterase activities in *OsPME1*-FOX and *OsPME34*-FOX. Pectin methylesterase activity is shown as the ratio of the activity in each FOX line inflorescence to that in the control. Error bars shows the SD (*n* = 3). Letters in each panel show significant differences at *p* < 0.01 (Tukey’s test).

**Figure 2 plants-12-01717-f002:**
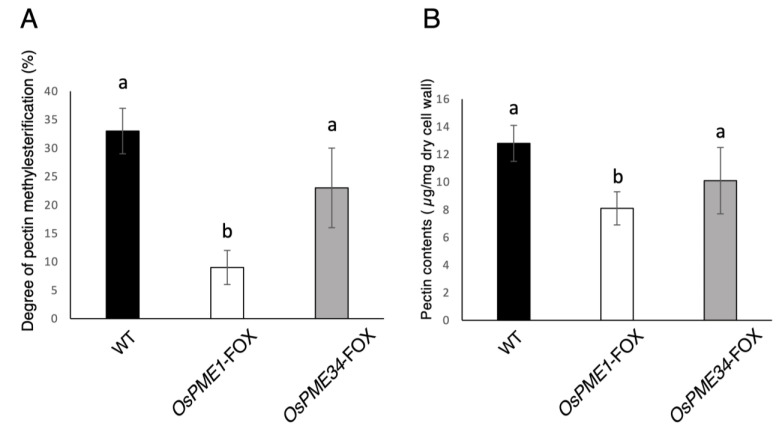
Pectin analysis of WT, *OsPME1*-FOX and *OsPME34*-FOX. (**A**) Degree of pectin methylesterification in flowers with anther length less than 1 mm from *OsPME1*-FOX and *OsPME34*-FOX. (**B**)The amounts of uronic acid from AIRs in the WT, *OsPME1-*FOX and *OsPME34-*FOX in flowers with anther length less than 1 mm. Error bars indicate the SD [*n* = 12 (FOX) and *n* = 4 (WT)]. Different letters in each panel indicate significant differences at *p* < 0.05 (Tukey’s test).

**Figure 3 plants-12-01717-f003:**
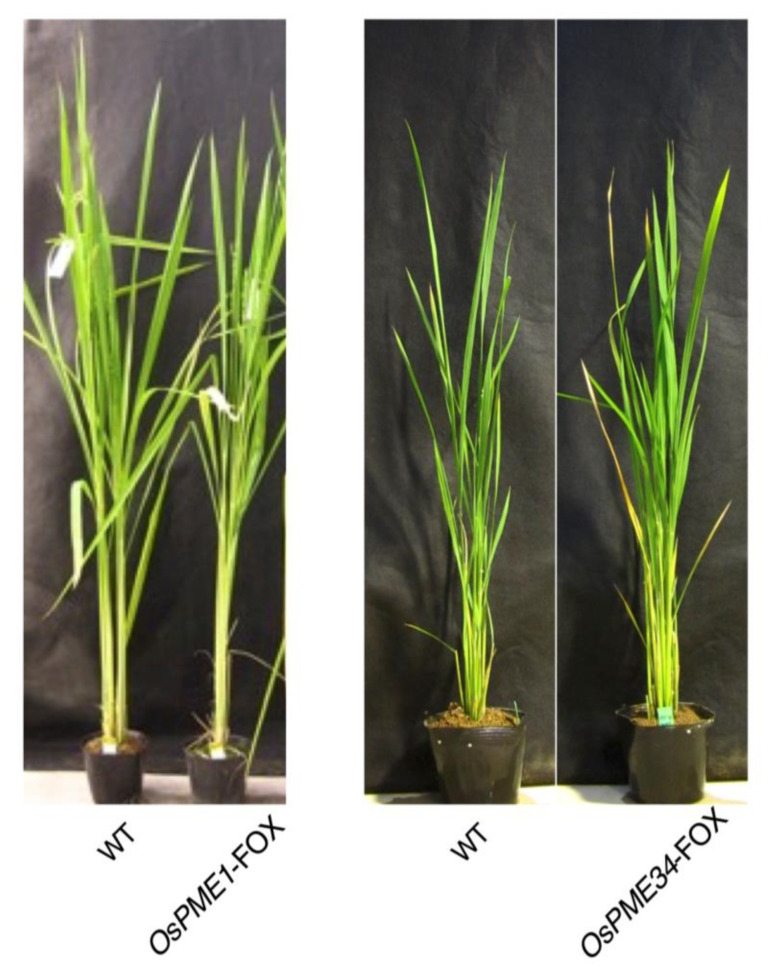
Vegetative phenotypes of *OsPME1*-FOX and *OsPME34*-FOX compared with WT plants). 60 days after seeding.

**Figure 4 plants-12-01717-f004:**
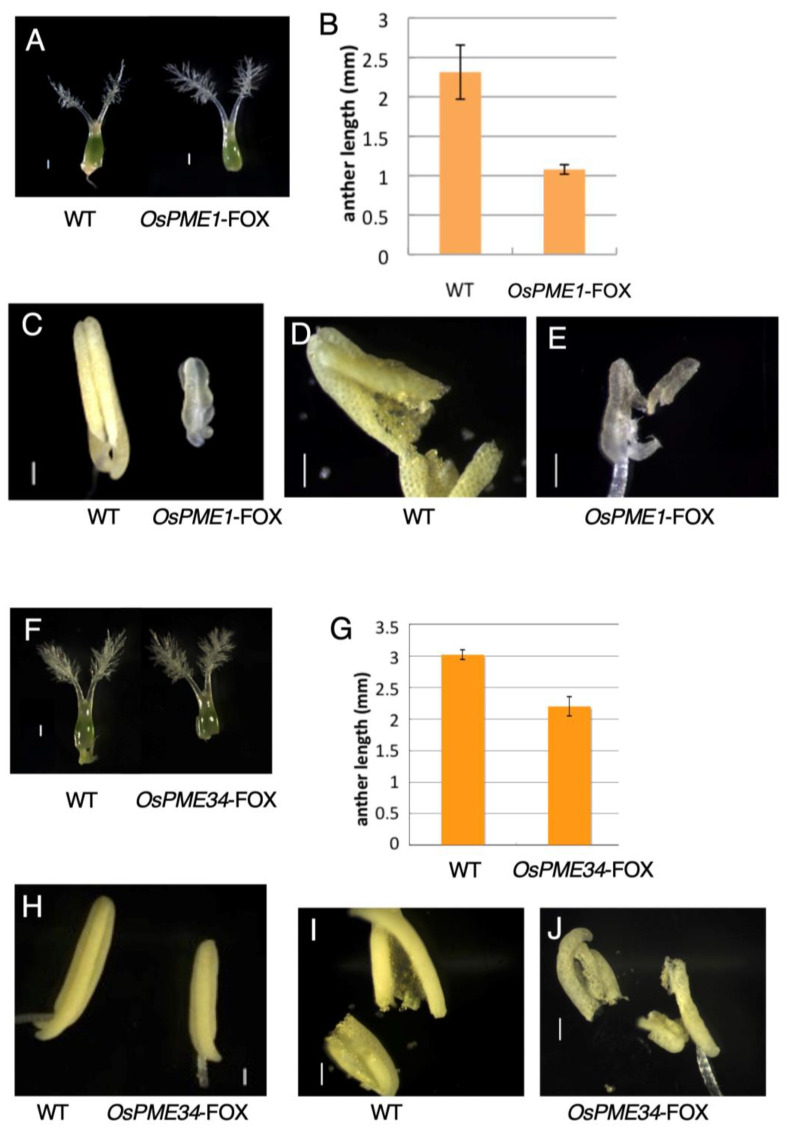
Reproductive phenotypes of *OsPME1*-FOX and *OsPME34*-FOX. Microscopic observation. (**A**,**F**) Mature pistils, (**B**,**G**) Mature anther length, (**C**,**H**) Mature anthers, (**D**,**I**) Torn anther of WT, (**E**) Torn anther of *OsPME1*-FOX, (**J**) Torn anther of *OsPME34*-FOX. Different letters in each panel indicate significant differences at *p*  <  0.05 (Tukey’s test). All experiment was biologically repeated at least four times independently with similar results. Scale bars = 0.1 mm.

**Figure 5 plants-12-01717-f005:**
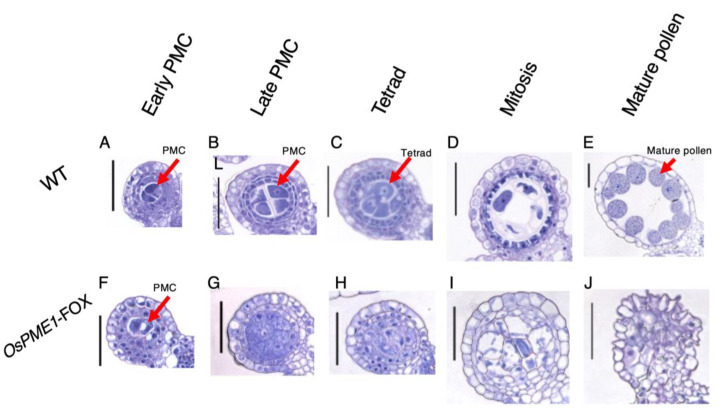
Abnormal pollen development of *OsPME1*-FOX. Sections were stained with toluidine blue O. (**A**–**E**; WT) and (**F**–**J**; *OsPME1*-FOX). All experiment was biologically repeated at least four times independently with similar results. PMC, pollen mother cell. Bars, 50 µm.

**Figure 6 plants-12-01717-f006:**
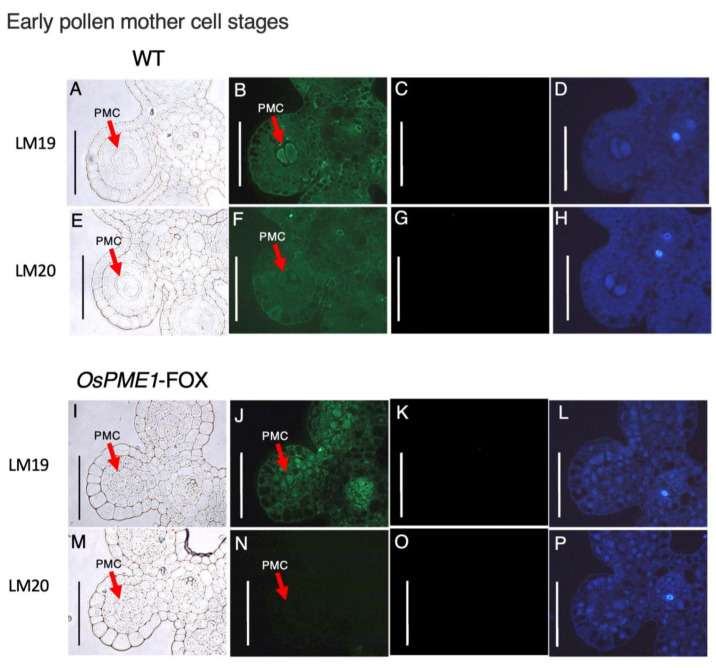
Immunohistochemistry of resin-embedded anther sections of early pollen mother-cell stages in WT and *OsPME1*-FOX. (**A**,**E**,**I**,**M**) Tissue cross-sections from the WT and *OsPME1*-FOX were observed under bright-field illumination. (**B**,**F**,**J**,**N**) Sections were stained with LM19 (anti-demethyl-esterified pectin) and LM20 (anti-methyl-esterified pectin) monoclonal antibodies. (**C**,**G**,**K**,**O**) The negative control, in which the first antibody step was omitted. (**D**,**H**,**L**,**P**) Autofluorescence excited by UV irradiation. All experiment was biologically repeated at least four times independently with similar results. PMC, pollen mother cell. Bars, 50 µm.

**Figure 7 plants-12-01717-f007:**
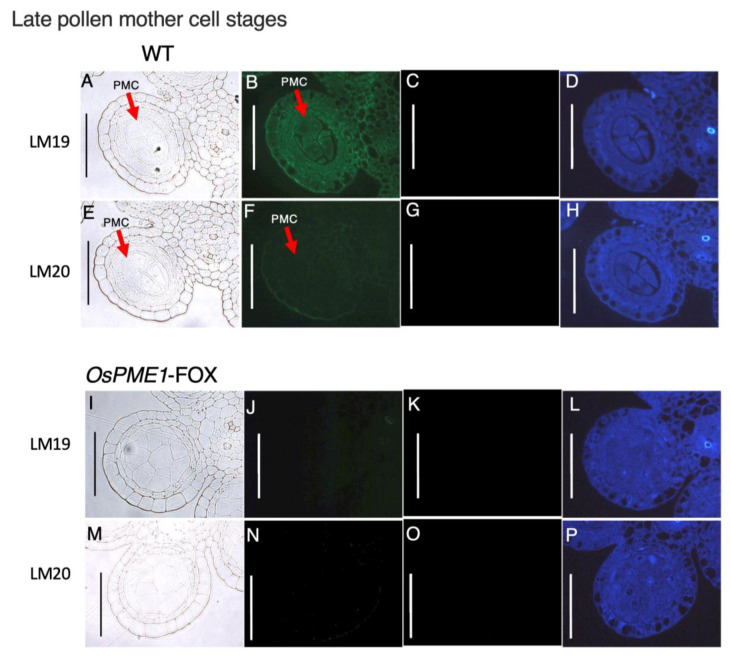
Immunohistochemistry of resin-embedded anther sections of late pollen mother-cell stages in WT and *OsPME1*-FOX. (**A**,**E**,**I**,**M**) Tissue cross-sections from the WT and *OsPME1*-FOX were observed under bright-field illumination. (**B**,**F**,**J**,**N**) Sections were stained with LM19 (anti-demethyl-esterified pectin) and LM20 (anti-methyl-esterified pectin) monoclonal antibodies. (**C**,**G**,**K**,**O**) The negative control, in which the first antibody step was omitted. (**D**,**H**,**L**,**P**) Autofluorescence excited by UV irradiation. All experiment was biologically repeated at least four times independently with similar results. PMC, pollen mother cell. Bars, 50 µm.

**Figure 8 plants-12-01717-f008:**
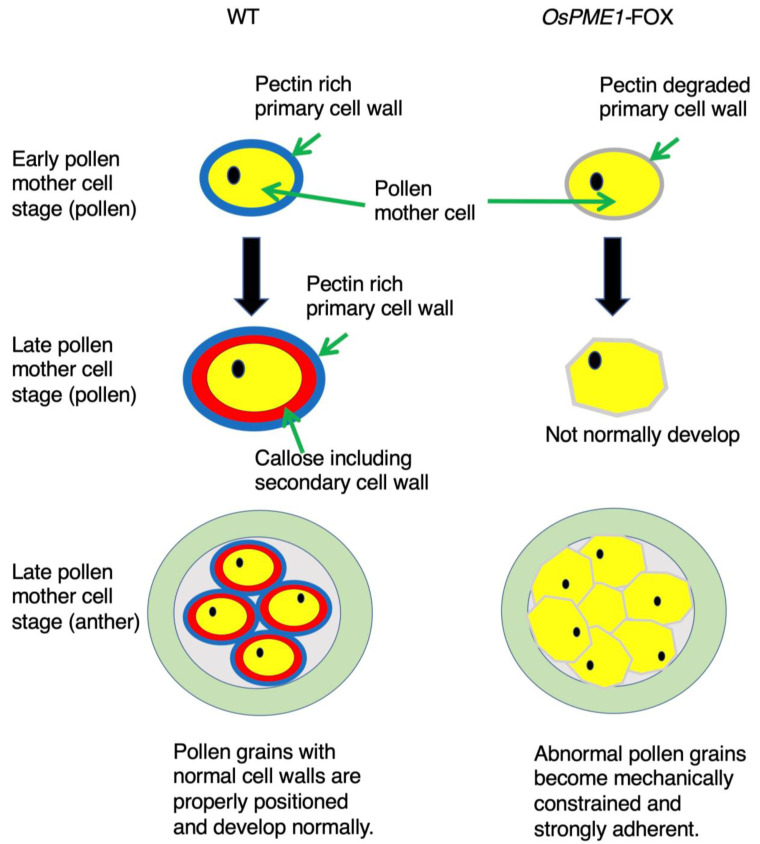
A model for Microspore Development in rice. The primary cell wall of the pollen mother cell has pectin. Callose is deposited between the primary wall and the plasma membrane of the pollen mother cell as the secondary cell wall. In *OsPME1*-FOX, the primary cell wall of the pollen mother cell is de-graded abnormally, which may prevent the normal development of microspores. In the anther, presumably causing the developing pollen grains to become mechanically constrained and remain strongly attached to the pollen grains.

## Data Availability

Not applicable.

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
