# Peer review of "Maintenance of Methyl-Esterified Pectin Level in Pollen Mother-Cell Stages Is Required for Microspore Development"

_plants, 2023, doi:10.3390/plants12081717_

Round 1

Reviewer 1 Report

This study is interesting and the experiments were well designed and performed. However, most of figures should be rearranged. For example, Figure 1 and Figure 4 were not well prepared. For Figure 5, it is hard to see the small figures of Figure 5A to Figure 5Jthus Figure 5 should be reorganized. In addition, the authors should add some arrows or letters to indicate the key part in Figure 5, Figure 6, and Figure 7.

Author Response

Dear Reviewer 1,

We are grateful to the editor and reviewers for their constructive and helpful comments, which greatly helped us improve our manuscript titled “Maintenance of Methyl Esterified Pectin Level in Pollen Mother Cell Stages is Required for Microspore Development” (plants-2272646). As indicated in the responses that follow, we have taken all these comments into account in preparing this resubmitted manuscript. Detailed responses to each comment are listed below.

Reviewer 1

This study is interesting and the experiments were well designed and performed. However, most of figures should be rearranged. For example, Figure 1 and Figure 4 were not well prepared. For Figure 5, it is hard to see the small figures of Figure 5A to Figure 5J,thus Figure 5 should be reorganized. In addition, the authors should add some arrows or letters to indicate the key part in Figure 5, Figure 6, and Figure 7.

Thank you so much for reviewer's advice. We rearranged all the figures. In Figure 5, we deleted the small figures. We added arrows to indicate the pollen mother cell in Figure 5, Figure 6, and Figure 7.

We believe our manuscript now meets standards for publication. I am looking forward to hearing from you soon.

Sincerely yours,

Hiroaki Iwai

University of Tsukuba, 

Institute of Life and Environmental Sciences, 

Tsukuba, Ibaraki 305-8572, Japan

iwai.hiroaki.gb@u.tsukuba.ac.jp

Reviewer 2 Report

The article deals with anther development mutants and shows that they have a decreased pectin distribution and content. The authors claim that pectin is the cause for anther developmental defects. The topic is interesting- however, I do not really understand the proposed mechanism- why less pectin in those mutants causes abnormal development? The functional explanation is not clear. It could be only correlative, the data does not show causation. Please expand on this and give a plausible mechanistic explanation.

 In addition, the editing is sloppy, with many unclear phrases, no italics, garbled titles etc. The article must be thoroughly edited.

 Here are some of my remarks:

 For introduction, please add a scheme/image of pollen structure, with different parts indicated. It’s very difficult to understand the text without it.

 L29-  “Later, in the tetrad stage, the cell wall is further degraded and the microspores are released into the meristem._ - I’m pretty much sure the spores are not released into meristem

 L35-“ a key procedure in the life cycle of angiosperms”- please change to “process”

L36- “ The main component of intine is composed of pectin, structural proteins [4], and microfibril cellulose” please change to “ Intine is composed of pectin, structural proteins [4], and microfibril cellulose”

  Arabidopsis thaliana, Nicotiana tabacum, E. coli etc- latin names should be in italics. Mutants should be in italics. Please fix this all over the manuscript, especially in the Introduction part.

 L56- you use the abbreviation PME for the first time without writing what the acronym means. The first time it’s mentioned must include both the abbreviated and the full form.

 L76 “The qrt1 mutant properly deposits [27] and degrades [28] secondary cell wall of the pollen mother cell is composed of callose, but the primary cell wall of the pollen mother cell remains partially intact after meiosis” – please edit this sentence

 L84-93 What is the hypothesis? What was the research question?

 M&M

Please transfer the Materials and Methods section before the results section

 At what stage did you sample for all the cell wall and molecular experiments? How did you chose those stages?

 Monitoring of Anther Development- How did you define the different stages of spikelet development for anatomy? What microtome did you use? How did you take the anther images- what microscope and camera?

 L338- Determination of Pectin Methylesterase Activity - How did you obtain those enzyme extracts from your tissue?

 L368 “All figures and tables should be cited in the main text as Figure 1, Table 1, etc.” please delete

 L371, Determination of the Pectin Methyl Ester Content- where did pectin solution come from? Which solution exactly is that? Please explain from which stage of the extraction protocol you took those solutions.

Results

Figures 4, 5,6,7- please check the letters. OsPME1-FOX appears in gibberish.

 Fig. 4, 5,6,7- how many biological repeats did you have in this experiment? Please write in the figure legend.

 Fig 5- you must write the parts you are writing about- such as tapetum, mother cells etc. Again I want to ask to add a schematic image of anther.

 Discussion

L265 -“It has been reported that pectin plays an important role in the reproductive organs of 265 rice, even though the cell wall is low in pectin in vegetative organs. Since the loss of the 266 pectin methyltransferase gene also had no significant effect on the growth of the vegetative organs and developmental defects were observed in the Pistil, a reproductive 268 organ, the regulation of pectin levels and methyl esterification may be an important function in the reproductive organs in rice. In the OsPME1-FOX phenotype, the decrease of 270 the degree of pectin methyl esterification level and deficient development in pollen deficient could be observed more stably than in OsPME34-FOX. OsPME1 has a conserved 272 PME domain and PMEI domain (group 2 PME), and OsPME34 only has PME domain 273 (group 1 PME). Group 2 PME may affects pollen development than group 1 PME.” 

It’s a very confusing paragraph. Please rewrite it and be consistent

Author Response

Dear Reviewer 2,

We are grateful to the editor and reviewers for their constructive and helpful comments, which greatly helped us improve our manuscript titled “Maintenance of Methyl Esterified Pectin Level in Pollen Mother Cell Stages is Required for Microspore Development” (plants-2272646). As indicated in the responses that follow, we have taken all these comments into account in preparing this resubmitted manuscript. Detailed responses to each comment are listed below.

Reviewer 2

The article deals with anther development mutants and shows that they have a decreased pectin distribution and content. The authors claim that pectin is the cause for anther developmental defects. The topic is interesting- however, I do not really understand the proposed mechanism- why less pectin in those mutants causes abnormal development? The functional explanation is not clear. It could be only correlative, the data does not show causation. Please expand on this and give a plausible mechanistic explanation. 

Thank you so much for reviewer's advice. We have added new Figure 8 and figure legend of Figure 8 on the structure of the pollen cell wall and the role of pollen mother cell wall pectin in pollen development to provide clarity for “Plants” readers.

Figure 8. A model for Microspore Development in rice. The primary cell wall of the pollen mother cell has pectin. Callose is deposited between the primary wall and the plasma membrane of the pollen mother cell as the secondary cell wall. In OsPME1-FOX, the primary cell wall of the pollen mother cell is de-graded abnormally, which may prevent the normal development of microspores. In the anther, presumably causing the developing pollen grains to become mechanically constrained and remain strongly attached to the pollen grains.

 In addition, the editing is sloppy, with many unclear phrases, no italics, garbled titles etc. The article must be thoroughly edited.

Excuse me for our many errors, we corrected the mistakes in the manuscripts.

 For introduction, please add a scheme/image of pollen structure, with different parts indicated. It’s very difficult to understand the text without it.

We have added new Figure S1 and figure legend on the structure of the pollen cell wall.

Figure S1. Model for microspore development in rice. Callose is deposited between the primary wall and the plasma membrane of the pollen mother cell. After the meiosis, callose is deposited between the microspores allowing synthesis of individual microspore walls. When the microspores initiate cell wall synthesis, the callose secondary wall and pectin primary wall of the pollen mother cell are degraded and microspores are released into the locule. After microspore release, exine is deposited on the microspores from the tapetum.

L29-  “Later, in the tetrad stage, the cell wall is further degraded and the microspores are released into the meristem._ - I’m pretty much sure the spores are not released into meristem

According to the reviewer’s comments, we corrected the mistakes of the sentences as follows.

Later, in the tetrad stage, the cell wall is further degraded and the microspores are released into the locule.

 L35-“ a key procedure in the life cycle of angiosperms”- please change to “process”

According to the reviewer’s comments, we corrected the mistakes of the sentences as follows.

a key process in the life cycle of angiosperms

L36- “ The main component of intine is composed of pectin, structural proteins [4], and microfibril cellulose” please change to “ Intine is composed of pectin, structural proteins [4], and microfibril cellulose”

According to the reviewer’s comments, we changed the sentence as follows.

L36

Intine is composed of pectin, structural proteins

  Arabidopsis thaliana, Nicotiana tabacum, E. coli etc- latin names should be in italics. Mutants should be in italics. Please fix this all over the manuscript, especially in the Introduction part.

We corrected the errors about italics in the manuscripts.

L40

The dex1 mutant of Arabidopsis thaliana exhibits

L75

In Arabidopsis, QUARTET1 (QRT1),

L78

which encodes polygalacturonase found in Arabidopsis,

L75

such as NtPPME1 from Nicotiana tabacum [24][25].

L78

when expressed in E. coli. [26].

 L56- you use the abbreviation PME for the first time without writing what the acronym means. The first time it’s mentioned must include both the abbreviated and the full form.

According to the reviewer’s advice, we use PME for the first time in L56, we included both the the abbreviated and the full form.

Thus, pectin methylesterase (PME) play

 L76 “The qrt1 mutant properly deposits [27] and degrades [28] secondary cell wall of the pollen mother cell is composed of callose, but the primary cell wall of the pollen mother cell remains partially intact after meiosis” – please edit this sentence

We changed the sentence as follows.

The qrt1 mutant properly deposits [27] and degrades [28] callose cell wall of the pollen mother cell is composed of callose, but the primary cell wall including pectin of the pollen mother cell remains after meiosis [28]. In WT, callose wall and pectin wall were properly deposits and degrades.

 L84-93 What is the hypothesis? What was the research question?

We transferred the L84-93 sentences to L62-70, and added the sentence as follows.

 M&M

Please transfer the Materials and Methods section before the results section

In the template file of the Instructions for Authors for "Plants", "Materials and Methods" is set fourth and is instructed to be after the discussion (https://www.mdpi.com/files/word-templates/plants-template.dot). Also, recent new articles have a similar format. Therefore, we have included "4. Materials and Methods" in this manuscript.

 At what stage did you sample for all the cell wall and molecular experiments? How did you chose those stages?

We added the sentence in Materials and Methods as follows.

L361

The developmental stages of rice pollen were defined in the same methods described as in the Itoh et al.[30] and Fujita et al.[14] based on anther length.

Figure 1. Characteristics of the OsPME1-FOX and OsPME34-FOX lines. (A) RT–PCR analysis of inflorescence transcripts from WT, OsPME1-FOX and OsPME34-FOX lines. The expression levels of OsPME1-FOX and OsPME34-FOX were higher than in the WT. The numbers in parentheses show the numbers of PCR cycles. These analyses were performed at least three times with similar results. (B) Pectin methylesterase activities in OsPME1-FOX and OsPME34-FOX. Pectin methylesterase activity is shown as the ratio of the activity in each FOX line inflorescence to that in the control. Error bars shows the SD (n  =  3). Letters in each panel show significant differences at P < 0.01 (Tukey’s test).

Figure 2. Pectin analysis of WT, OsPME1-FOX and OsPME34-FOX. (A) Degree of pectin methylesterification in flowers with anther length less than 1 mm from OsPME1-FOX and OsPME34-FOX. (B)The amounts of uronic acid from AIRs in the WT, OsPME1-FOX and OsPME34-FOX in flowers with anther length less than 1 mm.

 Monitoring of Anther Development- How did you define the different stages of spikelet development for anatomy? What microtome did you use? How did you take the anther images- what microscope and camera?

We added the sentence in Materials and Methods as follows.

L361

The developmental stages of rice pollen were defined in the same methods described as in the Itoh et al.[30] and Fujita et al.[14] based on anther length.

L365

Transverse sections of 3 µm were cut using a Leica VT1200S (Leica Microsystems, Wetzlar, Germany) and stained with 0.1% (w/v) toluidine blue O in distilled water (DW). The sections were visualized by microscopy (DMRB, Leica Microsystems, Wetzlar, Germany).

 L338- Determination of Pectin Methylesterase Activity - How did you obtain those enzyme extracts from your tissue?

We added the sentence as follows.

L390

Inflorescence including the anthers less than 1 mm in length from OsPME1-FOX and OsPME34-FOX were sampled.

 L368 “All figures and tables should be cited in the main text as Figure 1, Table 1, etc.” please delete.

We deleted the sentence.

 L371, Determination of the Pectin Methyl Ester Content- where did pectin solution come from? Which solution exactly is that? Please explain from which stage of the extraction protocol you took those solutions.

We added the sentence in “4.5. Extraction and Analysis of Cell Wall Polysaccharides” as follows.

L406

The dry cell walls were extracted with 50 mM Na2CO3 at room temperature for 2 h as pectic fraction.

We changed the words solution to fraction in “4.7. Determination of the Pectin Methyl Ester Content”.

L426

For hydrolysis of methyl esters bound to pectin, 0.1 m KOH (100 µl) was added to the pectin fraction (100 µg/100 µl),

Results

Figures 4, 5,6,7- please check the letters. OsPME1-FOX appears in gibberish.

We apologize for the gibberish figures. We have checked each figure and re-saved them all as image files (JPEG).

 Fig. 4, 5,6,7- how many biological repeats did you have in this experiment? Please write in the figure legend.

We added the sentence in the figure legend of Fig. 4, 5,6,7 as follows,

All experiment was biologically repeated at least four times independently with similar results.

 Fig 5- you must write the parts you are writing about- such as tapetum, mother cells etc. Again I want to ask to add a schematic image of anther.

Thank you so much for reviewer's advice. We rearranged all the figures. In Figure 5, we deleted the small figures. We added arrows to indicate the pollen mother cell in Figure 5, Figure 6, and Figure 7.

 Discussion

L265 -“It has been reported that pectin plays an important role in the reproductive organs of 265 rice, even though the cell wall is low in pectin in vegetative organs. Since the loss of the 266 pectin methyltransferase gene also had no significant effect on the growth of the vegetative organs and developmental defects were observed in the Pistil, a reproductive 268 organ, the regulation of pectin levels and methyl esterification may be an important function in the reproductive organs in rice. In the OsPME1-FOX phenotype, the decrease of 270 the degree of pectin methyl esterification level and deficient development in pollen deficient could be observed more stably than in OsPME34-FOX. OsPME1 has a conserved 272 PME domain and PMEI domain (group 2 PME), and OsPME34 only has PME domain 273 (group 1 PME). Group 2 PME may affects pollen development than group 1 PME.”  

It’s a very confusing paragraph. Please rewrite it and be consistent

Thank you for reviewer's advice, and we agree the comments. We have deleted the first half of the sentence and moved the second half of the sentence to L267.

We believe our manuscript now meets standards for publication. I am looking forward to hearing from you soon.

Sincerely yours,

Hiroaki Iwai

University of Tsukuba, 

Institute of Life and Environmental Sciences, 

Tsukuba, Ibaraki 305-8572, Japan

iwai.hiroaki.gb@u.tsukuba.ac.jp

Reviewer 3 Report

Dear Authors,

The manuscript by Hasegawa et al. “Maintenance of Methyl Esterified Pectin Level in Pollen Mother Cell Stages is Required for Microspore Development”, presents the role of homogalacturonan methyl esterification in pollen development in Oryza sativa. I think that in this form the manuscript needs to be improved. Unsatisfactory presentation of immunofluorescent data. The authors presented data with the localization of only two monoclonal antibodies, not covering the entire spectrum of possible modifications of homogalacturonan methyl esterification. Thus, LM19 represents a segment of the homogalacturonan molecule of four galacturonic acid residues without methyl groups, and LM20 of four galacturonic acid residues with methyl groups. But there are other variants of methyl esterification, which are not presented in the work. In addition, the immunofluorescence photographs themselves raise questions. In these photographs, the antibodies reveal the entirety of the cells (where they are shown) and should only be in the cell walls.

As minor remarks, it can be added that in the list of References according to the rules for authors, it is necessary to use lowercase letters in the title of articles.

Author Response

Dear Reviewer 3,

We are grateful to the editor and reviewers for their constructive and helpful comments, which greatly helped us improve our manuscript titled “Maintenance of Methyl Esterified Pectin Level in Pollen Mother Cell Stages is Required for Microspore Development” (plants-2272646). As indicated in the responses that follow, we have taken all these comments into account in preparing this resubmitted manuscript. Detailed responses to each comment are listed below.

Reviewer 3

The manuscript by Hasegawa et al. “Maintenance of Methyl Esterified Pectin Level in Pollen Mother Cell Stages is Required for Microspore Development”, presents the role of homogalacturonan methyl esterification in pollen development in Oryza sativa. I think that in this form the manuscript needs to be improved. Unsatisfactory presentation of immunofluorescent data. The authors presented data with the localization of only two monoclonal antibodies, not covering the entire spectrum of possible modifications of homogalacturonan methyl esterification. Thus, LM19 represents a segment of the homogalacturonan molecule of four galacturonic acid residues without methyl groups, and LM20 of four galacturonic acid residues with methyl groups. But there are other variants of methyl esterification, which are not presented in the work. In addition, the immunofluorescence photographs themselves raise questions. In these photographs, the antibodies reveal the entirety of the cells (where they are shown) and should only be in the cell walls.

We appreciate the reviewer's valuable comments. We have added the results of conventional ruthenium red pectin staining as a supplementary figure to support the significance of immunohistochemistry. According to reviewer’s comments, we have also added the following text to the results as follows.

L207

On the other hand, in anthers at the early pollen mother cell stage of OsPME1-FOX, the signal of de-methyl esterified pectin could be observed (Figure 6J), but the signal of methyl esterified pectin was not detected (Figure 6N). Although probably due to abnormal cell conditions, this signal does not detect only the cell wall and may be a nonspecific signal, staining of de-methyl esterified pectin and total pectin with ruthenium red (Figure S2) was very similar to the immunohistochemistry results (Figure 6).

Figure S2. Pectin staining in early pollen mother cell stages. De-methyl esterified pectin staining by ruthenium red of anther in early pollen mother cell stages, or all pectin staining, which treated with 0.1 M NaOH prior to staining to de-esterify pectin. The experiment was repeated at least four times independently with similar results. Bars 50 µm.

As minor remarks, it can be added that in the list of References according to the rules for authors, it is necessary to use lowercase letters in the title of articles.

Thank you for reviewer’s advice. We checked the all references and used the “Zotero” soft, according to Instructions for Authors “The bibliography software package EndNoteZoteroMendeleyReference Manager are recommended.”.

We believe our manuscript now meets standards for publication. I am looking forward to hearing from you soon.

Sincerely yours,

Hiroaki Iwai

University of Tsukuba, 

Institute of Life and Environmental Sciences, 

Tsukuba, Ibaraki 305-8572, Japan

iwai.hiroaki.gb@u.tsukuba.ac.jp

Round 2

Reviewer 2 Report

The article is much improved and looks fine to me. Good luck!

Author Response

Dear Reviewer 2,

We are grateful to the reviewer for the helpful comments, which helped us improve our manuscript titled “Maintenance of Methyl Esterified Pectin Level in Pollen Mother Cell Stages is Required for Microspore Development” (plants-2272646).

Reviewer 2

The article is much improved and good luck!

Thank you so much for your kind response.

We have checked English quality and spelling throughout the paper.

We believe our manuscript now meets the standards for publication.

Sincerely yours,

Hiroaki Iwai

Reviewer 3 Report

Dear Authors,

Indeed, ruthenium red is often used to detect pectin compounds. However, this staining is not specific. Ruthenium red has long been used in light microscopy to stain pectin, plant mucins, cellulose, starch, inulin and lignin, nucleic acids, and other substances. Back in 1971, it was written that ruthenium red «is found to stain in addition α-, β- and γ-oxycelluloses, hemicelluloses, gums, galactans, free lignin, mannan, and amylohemicelluloses» (Luft, J. H. (1971). Ruthenium red and violet. I. Chemistry, purification, methods of use for electron microscopy and mechanism of action. The Anatomical Record, 171(3), 347–368), and in 1984 – «Because of its high positive charge, RR (ruthenium red) can react only not with acidic glycosaminoglycans» but also with other polyanions, e.g. nucleic acids (Gutierrez-Gonzalvez MG, Stockert JC, Ferrer JM, Tato A. Ruthenium red staining of polyanion containing structures in sections from epoxy-resin embedded tissues. Acta Histochem. 1984; 74(1): 115-120.). Therefore, in a manuscript discussing the ratio of methylated and demethylated homogalacturonan, the use of this staining is inadequate. In addition, the manuscript does not describe the method used for staining with ruthenium red.

I insist that the selected microscopic methods and the antibodies used do not reveal the full picture of changes in the modification of cell wall homogalacturonans in plants with overexpression of pectin methylesterase genes. Moreover, the complete absence of methyl esterified homogalacturonan in anther cells is doubtful, since this modification of homogalacturonan is a synthesized form and should be observed in vesicles of the Golgi apparatus and transport vesicles, and pectin methylesterases work exclusively in cell walls.

Author Response

Dear Reviewer 3,             

We are grateful to the reviewer for the helpful comments, which helped us improve our manuscript titled “Maintenance of Methyl Esterified Pectin Level in Pollen Mother Cell Stages is Required for Microspore Development” (plants-2272646).

Reviewer 3

Indeed, ruthenium red is often used to detect pectin compounds. However, this staining is not specific. Ruthenium red has long been used in light microscopy to stain pectin, plant mucins, cellulose, starch, inulin and lignin, nucleic acids, and other substances. Back in 1971, it was written that ruthenium red «is found to stain in addition α-, β- and γ-oxycelluloses, hemicelluloses, gums, galactans, free lignin, mannan, and amylohemicelluloses» (Luft, J. H. (1971). Ruthenium red and violet. I. Chemistry, purification, methods of use for electron microscopy and mechanism of action. The Anatomical Record, 171(3), 347–368), and in 1984 – «Because of its high positive charge, RR (ruthenium red) can react only not with acidic glycosaminoglycans» but also with other polyanions, e.g. nucleic acids (Gutierrez-Gonzalvez MG, Stockert JC, Ferrer JM, Tato A. Ruthenium red staining of polyanion containing structures in sections from epoxy-resin embedded tissues. Acta Histochem. 1984; 74(1): 115- 120.). Therefore, in a manuscript discussing the ratio of methylated and demethylated homogalacturonan, the use of this staining is inadequate. In addition, the manuscript does not describe the method used for staining with ruthenium red.

We deleted the Figure S2 about ruthenium red staining, and the sentences.

L97

and ruthenium red

L239

Although probably due to abnormal cell conditions, this signal does not detect only the cell wall and may be a nonspecific signal, staining of de-methyl esterified pectin and total pectin with ruthenium red (Figure S2) was very similar to the immunohistochemistry results (Figure 6).

I insist that the selected microscopic methods and the antibodies used do not reveal the full picture of changes in the modification of cell wall homogalacturonans in plants with overexpression of pectin methylesterase genes. Moreover, the complete absence of methyl esterified homogalacturonan in anther cells is doubtful, since this modification of homogalacturonan is a synthesized form and should be observed in vesicles of the Golgi apparatus and transport vesicles, and pectin methylesterases work exclusively in cell walls.

Thank you for your comments.

The LM19 (de-methyl esterified pectin-specific antibody) and LM20 (methyl esterified -specific antibody) used in our paper were established by Verhertbruggen Y et al. This method has been established by at least 200 papers published to date using the same method as our paper.

In addition, this antibody is commercially available and used by many researchers.

https://filgen.jp/Product/Bioscience4/PlantProbes/

Verhertbruggen Y, Marcus SE, Haeger A, Ordaz-Ortiz JJ, Knox JP. An extended set of monoclonal antibodies to pectic homogalacturonan. Carbohydr Res. 2009 Sep 28;344(14):1858-62. doi: 10.1016/j.carres.2008.11.010. Epub 2008 Nov 27. PMID: 19144326.

To respond to revewer3's point, we need biochemical results of the pollen mother cell wall, but such an analysis is not possible because pollen mother cells are very small and it is very difficult to collect only pollen mother cells. We believe that immunohistochemical staining is the most effective method to reach the objective of this paper.

The distribution in the Golgi can be reliably observed in immunohistochemistry using LM19 and LM20 antibodies when immunogold electron microscopy is used, and there are no examples observed in experiments using fluorescence microscopy. This may be because the exposure time is the same for WT and OsPME1-FOX, and the amount of pectin itself is much higher in the cell wall than in the Golgi, and therefore Golgi body signals are very difficult to be observed. However, likely, the results of our experiment do not deviate from the results of previous papers, since it would be extremely difficult to stain and observe pectin in the Golgi apparatus in microspores under a fluorescence microscope, based on previous results.

For example,

https://www.frontiersin.org/articles/10.3389/fpls.2021.703713/full#B78

https://www.frontiersin.org/articles/10.3389/fpls.2021.703713/full

https://www.mdpi.com/1422-0067/21/14/4840

https://www.ncbi.nlm.nih.gov/pmc/articles/PMC281612/

We added the sentences as follows,

L239

Although methyl esterified pectin is also included in the Golgi vesicles, these experiments were focused on the cell wall, exposure times are the same in WT and OsPME1-FOX, and the amount of pectin itself is much higher in the cell wall than in the Golgi vesicles, and therefore methyl esterified pectin signals in Golgi vesicle are very difficult to be observed.

The comments of reviewer 3 said while this paper may not reveal the full extent of the changes by immunohistochemistry using LM19 and LM20 in pollen mother cell wall homogalacturonan modification in pectin methylesterase overexpressor, it is important to note that overexpression of PME results in aberrant pollen mother cell walls and aborted pollen formation. We believe this paper contribute the understand the microspore development, and will provide Plants readers with important new information for the cell wall in pollen mother cell.

We believe our manuscript now meets standards for publication. I am looking forward to hearing from you soon.

Sincerely yours,

Hiroaki Iwai

University of Tsukuba,
Institute of Life and Environmental Sciences, Tsukuba, Ibaraki 305-8572, Japan iwai.hiroaki.gb@u.tsukuba.ac.jp
